# Pharmacokinetic/Pharmacodynamic Analysis and Dose Optimization of Cefmetazole and Flomoxef against Extended-Spectrum β-Lactamase-Producing *Enterobacterales* in Patients with Invasive Urinary Tract Infection Considering Renal Function

**DOI:** 10.3390/antibiotics11040456

**Published:** 2022-03-28

**Authors:** Yukihiro Hamada, Hidefumi Kasai, Moeko Suzuki-Ito, Yasufumi Matsumura, Yohei Doi, Kayoko Hayakawa

**Affiliations:** 1Department of Pharmacy, Tokyo Women’s Medical University Hospital, Tokyo 162-8666, Japan; suzuki.moeko@twmu.ac.jp; 2School of Medicine, Keio University, Tokyo 160-8582, Japan; hidefumi.kasai@nifty.com; 3Department of Clinical Laboratory Medicine, Kyoto University Graduate School of Medicine, Kyoto 606-8507, Japan; yazblood@kuhp.kyoto-u.ac.jp; 4Center for Innovative Antimicrobial Therapy, Division of Infectious Diseases, University of Pittsburgh School of Medicine, Pittsburgh, PA 15261, USA; yoheidoi@gmail.com; 5Department of Microbiology and Infectious Diseases, Fujita Health University School of Medicine, Aichi 470-1192, Japan; 6Disease Control and Prevention Center, National Center for Global Health and Medicine, Tokyo 162-8655, Japan; khayakawa@hosp.ncgm.go.jp

**Keywords:** antimicrobial stewardship, cefmetazole, flomoxef, pharmacokinetics/pharmacodynamics, Monte Carlo simulations

## Abstract

The optimal regimens of cefmetazole and flomoxef for the treatment of urinary tract infections caused by extended-spectrum β-lactamase (ESBL)-producing *Enterobacterales* are not well defined. Our study found that the pharmacokinetic/pharmacodynamic targets for cefmetazole and flomoxef were 70% T > MIC, which is suggestive of bactericidal activity. A Monte Carlo simulation (MCS) was performed using the published data to calculate a new probability of target attainment (PTA ≥ 90%) for each renal function. The MCS was performed with 1000 replicates, and clinical breakpoints were calculated to attain PTA ≥ 90% for creatinine clearance (CCR) of 10, 30, 50, and 70 mL/min. The 90% ≥ PTA (70% T > MIC) of cefmetazole and flomoxef in patients who received a standard regimen (0.5 or 1 g, 1 h injection) for each renal function was calculated. Our results suggest that in patients with CCR of less than 30, 31–59, and more than 60 mL/min, the optimal dosage of cefmetazole would be 1 g q12 h, 1 g q8 h, and 1 g q6 h, respectively. Furthermore, in patients with CCR of less than 10, 10–50, and more than 50 mL/min, the optimal dosage of flomoxef would be 1 g q24 h, 1 g q8 h or 12 h, and 1 g q6 h, respectively.

## 1. Introduction

Abuse of broad-spectrum antibiotics is one of the major causes of the development of antimicrobial-resistant bacteria. The problem of antimicrobial resistance has become a public threat [1]. The frequency of occurrence of urinary tract infections (UTIs) caused by extended-spectrum β-lactamase (ESBL)-producing *Enterobacterales* has been increasing globally. As a top healthcare priority, the World Health Organization declared the development of new antibiotics active against ESBL-producing *Enterobacterales* in 2017 [2]. Among ESBL-producing *Enterobacterales*, ESBL-producing *Escherichia coli* (ESBL-*E. coli*) is considered the greatest threat [3,4]. The number of patients infected with it is increasing worldwide [5,6], especially in Africa, Latin America, and Asia [7]. Carbapenems are often the drug of choice for the treatment of severe infections due to ESBL-producing *Enterobacterales*. However, the excessive use of carbapenems in such cases promotes carbapenem resistance [8,9]. The RCT Meropenem vs. Piperacillin-Tazobactam for Definitive Treatment of BSI’s Due to Ceftriaxone Non-susceptible *Escherichia Coli* and *Klebsiella* spp. (MERINO) study revealed that piperacillin–tazobactam should be avoided for the targeted therapy of bloodstream infections caused by ESBL-producing *E. coli* and *K. pneumoniae* [10]. However, there remains a possibility that the treatment options for non-bacteremic UTIs caused by ESBL have not yet been sufficiently validated. Cefmetazole, a cephamycin agent, is stable against hydrolysis by ESBL; therefore, it has strong in vitro activity against ESBL-producing *Enterobacterales* at low minimum inhibitory concentrations (MICs) [11]. Flomoxef is a beta-lactam antibiotic with oxygen substituted for sulfur and 7-α-methoxy group in the cephalosporin core. Like cefmetazole, flomoxef has been reported to possess high antibacterial activity against ESBL-producing *Enterobacterales* in in vitro studies [12,13]. Previous studies have shown that cefmetazole and flomoxef have therapeutic efficacy against ESBL-producing *E. coli* infections that is comparable to carbapenems [14,15,16]. There are few clinical data on the potential value of cefmetazole and flomoxef for the treatment of ESBL-associated infections [14,17,18,19]. In addition, there have been few studies on the appropriate doses or validated breakpoints for this indication [20]. Hence, we conducted a retrospective observational study to determine optimal doses of cefmetazole and flomoxef that correlate with clinical efficacy [21]. The purpose of this study was to evaluate the appropriate dose and clinical pharmacokinetic/pharmacodynamic (PK/PD) breakpoints of cefmetazole and flomoxef for UTIs caused by ESBL-producing *Enterobacterales* considering renal function.

## 2. Results

Probability of Target Attainment (PTA) for Cefmetazole and Flomoxef Based on Renal Function

The probability of target attainment (PTA) was calculated by simulation using dosage, renal function, and MIC for cefmetazole and flomoxef, respectively. Table 1 was used for PK parameters [20,22]. The PTA was calculated as the probability of 70% T > MIC and the results are shown in Figure 1.

Figure 1A shows the PTA for cefmetazole in patients who received the following standard regimens for each renal function in Japan. At a creatinine clearance (CCR) of 10 mL/min, the standard regimen of cefmetazole (0.5 or 1 g/24 h, 1 h infusion) could achieve a PTA for cefmetazole (70% T > MIC) of >90% at an MIC of 8 mg/L. At a CCR of 30 mL/min, the standard regimen of cefmetazole (0.5 or 1 g/12 h, 1 h infusion) could achieve a PTA for cefmetazole (70% T > MIC) of >90% at an MIC of 4 mg/L. Moreover, at a CCR of 50 mL/min, the standard regimen of cefmetazole (0.5 or 1 g/8 or 12 h, 1 h infusion) could achieve a PTA for cefmetazole (70% T > MIC) of >90% at MICs from 1 to 4 mg/L. At a CCR of 70 mL/min, the standard regimen of cefmetazole (0.5 or 1 g/6 or 8 h, 1 h infusion) could achieve a PTA for cefmetazole (70% T > MIC) of ≥90% at MICs from 1 to 4 mg/L.

Figure 1B shows the PTA for flomoxef in patients who received standard regimens for each renal function. At a CCR of 10 mL/min, the standard regimen of flomoxef (0.5 or 1 g/12 h, 1 h infusion) could achieve a PTA for flomoxef (70% T > MIC) of ≥90% at an MIC of 8 mg/L. At a CCR of 30 mL/min, the standard regimen of flomoxef (0.5 or 1 g/8 or 12 h, 1 h infusion) could achieve a PTA for flomoxef (70% T > MIC) of ≥90% at MICs from 0.5 to 4 mg/L At a CCR of 50 mL/min and the standard regimen of flomoxef (0.5 or 1 g/6 or 8 h, 1 h infusion) could achieve a PTA for flomoxef (70% T > MIC) of ≥90% at MICs from 0.5 to 2 mg/L. At a CCR of 70 mL/min, the standard regimen of flomoxef (0.5 or 1 g/6 or 8 h, 1 h infusion) could achieve a PTA for flomoxef (70% T > MIC) of ≥90% at MICs from 0.0625 to 0.25 mg/L. Table 2 shows the PK/PD breakpoints of cefmetazole and flomoxef standard regimens at different renal functions with PTA ≥ 90%.

## 3. Discussion

To the best of our knowledge, this is the first study to evaluate the appropriate dosing of cefmetazole and flomoxef based on renal function for UTIs caused by ESBL-producing *Enterobacterales*. Cefmetazole and flomoxef are gaining increasing attention as potential carbapenem-sparing treatment options for infections caused by ESBL-producing *Enterobacterales*, and they are commercially available and commonly used in Japan [14]. Concerns have been raised over their proper use to maintain their effectiveness and prevent the emergence of resistance. Pathophysiological and clinical factors associated with UTIs can affect the pharmacokinetics profile of antibiotics. Therefore, inadequate dosing regimens could lead to treatment failures, increased emergence of resistance, and higher mortality rates.

Matsumura et al. reported on the efficacy of cefmetazole, flomoxef, and carbapenems for the treatment of ESBL-*E. coli* bacteremia and found that in non-immunocompromised patients, cefmetazole or flomoxef therapy of ESBL-*E. coli* bacteremia was not inferior to carbapenem therapy in terms of mortality [14]. In their study, the majority (>90%) of the patients received cefmetazole at 1 g every 8 h and flomoxef at 1 g every 8 h (or adjusted equivalent doses for renal dysfunction). However, the information on time above minimal inhibitory concentration (TAM), a PK/PD parameter, was not available. We therefore calculated TAM with the total cefmetazole concentration without protein binding (about 85% or less) [23]. To significantly affect free drug levels, the protein binding should be greater than 80% based on PK parameter considerations [24]. However, the greater intrinsic activity of lipophilic drug allows for the compensation of extensive protein binding [25]. During excretion, cefmetazole is more highly concentrated in the urine than in the plasma, which explains its effectiveness against UTIs [26]. Animal model studies suggest that the PD target associated with efficacy in the treatment of ESBL-producing *Enterobacterales* infections are the equivalent TAM used in non-ESBL-producing organisms [27]. In our previous study to evaluate the appropriate dosing of cefmetazole, TAM was relatively high as total concentration was used without considering protein binding [21]. In that study, the TAM of patients who did not respond to cefmetazole was 65.8%; cefmetazole was clinically efficacious in all five patients with TAM less than 50%. This result suggests that the host factor may be more important than TAM in treatment failure. [21]. Therefore, we concluded that this result, which was based on total concentration without considering protein binding, was reasonable. Tashiro et al. [28] reported that free T > MIC is the most significant PK/PD index of flomoxef against ESBL-*E. coli* and its target value is greater than 40%. The protein binding of flomoxef was 36.2 ± 0.5% [29]. Hence, the use of target TAM of 70% using total concentration is considered valid even if the free concentration is approximately 60%.

In the aforementioned study, the MIC_50_ and MIC_90_ of 121 ESBL-*E. coli* isolates were ≤1 and 2 mg/L for cefmetazole and ≤1 and ≤1 mg/L for flomoxef, respectively [14]. According to another study that evaluated the antimicrobial susceptibility of pathogens isolated from surgical site infections in Japan, MIC_90_ and MIC_50_ for 41 ESBL-producing *Enterobacterales*, of which 35 were ESBL-*E. coli*, were as follows: MIC_90_ and MIC_50_ for cefmetazole were 8 mg/L and 1 mg/L and MIC_90_ and MIC_50_ for flomoxef were 1 mg/L and <0.063 mg/L, respectively [30]. Although a large number of data are needed to accurately determine the susceptibility of ESBL-*E. coli* in Japan, at least based on these reports, a majority of ESBL-*E. coli* isolates seem to have MICs lower than or equal to the PK/PD breakpoints identified in this study, based on these reports.

This study has several limitations. First, the simulation was performed without considering the protein binding, since we did not measure the actual free concentration. In our previous study, the median calculated TAM was 92.6%, which was relatively high since the total cefmetazole concentration without considering the protein binding was used. Its clinical effectiveness was more than 90%, supporting this result [21]. Second, the PK parameters of the volume of distribution (Vd) and infusion time of 1 h were fixed, implying that the dosage should be increased for overweight patients. Nakai et al. [31] reported that ESBL E. coli is the problem for both nosocomial and community-acquired infections in Japan. Although, logistically challenging in the outpatient setting, further prolongation of the infusion time is likely to improve the clinical effectiveness [21]. Continuous or prolonged infusion may represent the best administration choice for maximizing the pharmacodynamics of beta-lactams under the same daily dose. Improved attainment of a certain PK/PD threshold with continuous infusion compared to intermittent infusion may show remarkable benefits in severe beta-lactam infection or under augmented renal clearance [32]. To maximize the use of existing antimicrobial agents, further interventions should include prolonged/continuous infusion into practice [33]. Although MCS is a useful tool for determining appropriate empirical antibiotic dosage regimens, clinical trials are needed to validate the efficacy and safety of higher dosages and extended or continuous infusions. The development of long-acting beta-lactam agents with activity against ESBL-producing *Enterobacterales* is also desirable in this regard.

## 4. Materials and Methods

### 4.1. Pharmacokinetics Parameters

The PK parameters of cefmetazole were calculated using the results reported by Tomizawa et al. [34], where a one-compartment model was used in which clearance was related to CCR and Vd was weight-dependent. The CCR was calculated using the Cockcroft–Gault equation [20]. In this study, the body weight was fixed at 60 kg. The PK of flomoxef was simulated using the 2-compartment model reported by Ito et al. [20]. Although significant covariates were not reported in this model, the effects of renal function on flomoxef clearance were considered using the following data and assumptions [35]. We calculated the half-life of flomoxef for several CCR (Table 3), by which we predicted flomoxef total CL for each CCR with the assumption that CL was inversely proportional to the half-life. Although inter-individual variabilities for flomoxef were not reported in [36], we assumed the same variability of CL as that of cefmetazole (20%). In addition, variabilities for the other three parameters were also set to 20%.

### 4.2. Pharmacodynamics Data

*Enterobacterales* (*E. coli*, *K. pneumoniae, Klebsiella oxytoca*, and *Proteus mirabilis*) are the most common ESBL-producing pathogens of GNB causing infections [11]. Several MICs of cefmetazole and flomoxef for *Enterobacterales* were evaluated by fixing them in the range of 0.0625 to 128 mg/L.

### 4.3. Pharmacokinetic/Pharmacodynamic Target and Analysis

The PK/PD targets for cefmetazole and flomoxef were 70% T > MIC, which shows bactericidal activity [36,37]. In our previous clinical study [21], it was confirmed that the clinical efficacy was ≥90% when T > MIC with ≥70%, hence we decided to perform the simulation with this target value. Phoenix NLME version 8.1 (Certara, Princeton, NJ, USA) was utilized for PK and Monte Carlo simulation (MCS) and R version 3.3.2 was used to calculate time above MIC. A MCS study was performed using the published data to calculate the probability of target attainment (PTA ≥ 90%) by simulation [20]. In the current study, MCSs with 1000 replicates were performed using the PK parameters listed in Table 1. The clinical breakpoints were calculated to attain PTA ≥ 90% for creatinine clearance (CCR) of 10, 30, 50, and 70 mL/min, respectively. The 90% ≥ PTA (70% T > MIC) of cefmetazole and flomoxef in patients who received a standard regimen (0.5 or 1 g, 1 h injection) for each renal function was calculated.

## 5. Conclusions

To our knowledge, appropriate doses of these drugs for ESBL-producing *Enterobacterales* have not been studied according to renal function. Clinical implementation of PK/PD theory can play a critical role in controlling AMR. Our findings serve as a foundation for future clinical studies that address the utility of cefmetazole and flomoxef as carbapenem-sparing treatment options for UTIs caused by ESBL-producing *Enterobacterales*. Our results suggest that in patients with CCRs of less than 30, 31–59, and more than 60 mL/min, the optimal dosage of cefmetazole would be 1 g q12, 1 g q8, and 1 g q6, respectively, and in patients with CCRs of less than 10, 10–50, and more than 50 mL/min, the optimal dosage of flomoxef would be 1 g q24, 1 g q8 or 12, and 1 g q6, respectively.

## Figures and Tables

**Figure 1 antibiotics-11-00456-f001:**
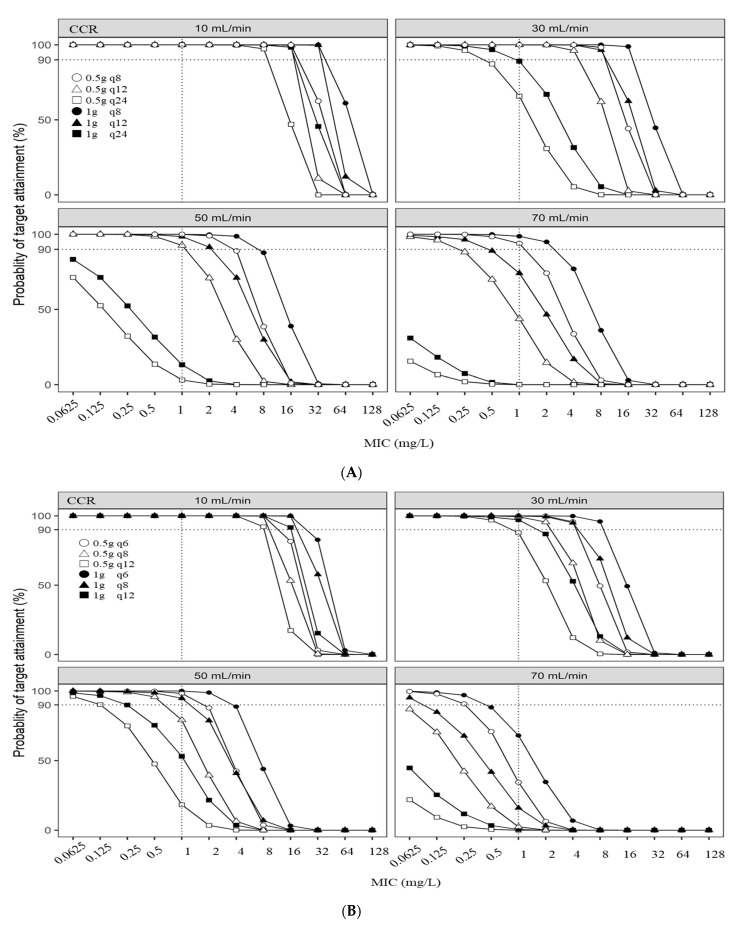
Probabilities of target attainment for cefmetazole (**A**) and flomoxef (**B**) doses with an infusion duration of 1 h were simulated. The model simulated cefmetazole and flomoxef clearance as a function of creatinine clearance (CCR) within four categories of estimated renal function: 10, 30, 50, and 70 mL/min. The PTA was benchmarked on 70% cefmetazole and flomoxef concentration time above the MIC (70% T > MIC).

**Table 1 antibiotics-11-00456-t001:** Population pharmacokinetic models for Monte Carlo simulation [20,22].

**Cefmetazole**	**1-Compartment Model**	**Final Model**	
		Pharmacokinetic Parameters	
			CL (L/h) = 0.0704 × CCR
			Vd (L) = 0.163 × BW
		Variability	
			ωCL (%) = 21.0
			ωVd (%) = 8.4
			σ (%) = 13.5
**Flomoxef**	**2-Compartment Model**	**Final Model**	
		Pharmacokinetic Parameters	
			Vc (L) = 7.14
			K_10_ (h^−1^) = 2.12
			K_12_ (h^−1^) = 2.45
			K_21_ (h^−1^) = 2.57
		Variability	
			ωVc (%) = 20.0
			ωK_10_ (%) = 20.0
			ωK_12_ (%) = 20.0
			ωK_21_ (%) = 20.0

CCR: creatinine clearance (mL/min), BW: body weight (kg), ω: inter-individual variability, σ: intra-individual variability, Vd and Vc: volume distribution for the total (1-compartment model) or for central compartment (2-compartment model).

**Table 2 antibiotics-11-00456-t002:** Clinical pharmacokinetic/pharmacodynamic (PK/PD) breakpoints of UTIs caused by ESBLs for cefmetazole and flomoxef with PTA ≥ 90% for each renal function.

**Dose (1 h Infusion)**		**Cefmetazole PK/PD Breakpoint (mg/L)**			
**mL/min**		**CCR 10**	**CCR30**	**CCR 50**	**CCR70**
500 mg	q12	16	4	1	0.125
1000 mg		32	8	2	0.25
500 mg	q8	16	8	2	1
1000 mg		32	16	4	2
500 mg	q6	32	16	4	2
1000 mg		64	32	8	4
**Dose (1 h Infusion)**		**Flomoxef PK/PD Breakpoint (mg/L)**			
**mL/min**		**CCR 10**	**CCR30**	**CCR 50**	**CCR70**
500 mg	q12	8	0.5	0.125	<0.0625
1000 mg		16	1	0.125	<0.0625
500 mg	q8	8	2	0.5	<0.0625
1000 mg		16	4	1	0.0625
500 mg	q6	8	4	1	0.25
1000 mg		16	8	2	0.25

CCR: creatinine clearance (mL/min).

**Table 3 antibiotics-11-00456-t003:** Pharmacokinetics parameters of flomoxef.

**Pharmacokinetics Parameter Calculation Formula for Flomoxef by Renal Function; as a 1 h Infusion of Flomoxef 1 g**				
		**n**	**T_1/2_ (β) (h)**	**Model**
**Healthy**		25	0.82	CL (L/h) = Vc × K_10_
**Renal dysfunction**				**CL conversion formula**
	5 ≦ CCR ≦ 20	4	6.95	CL severe = CL healthy × (1/0.82)/6.95
	(severe)			
	20 < CCR ≦ 40	10	2.48	CL mild = CL healthy × (1/0.82)/2.48
	(mild)			
	40 < CCR ≦ 70	10	1.57	CL medium = CL healthy × (1/0.82)/1.57
	(medium)			

CCR: creatinine clearance (mL/min), Vc: total volume distribution (1-compartment model).

## Data Availability

All applicable data are contained in the paper.

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
