# Peer review of "Pharmacokinetic/Pharmacodynamic Analysis and Dose Optimization of Cefmetazole and Flomoxef against Extended-Spectrum β-Lactamase-Producing Enterobacterales in Patients with Invasive Urinary Tract Infection Considering Renal Function"

_antibiotics, 2022, doi:10.3390/antibiotics11040456_

Round 1
Reviewer 1 Report
The article submitted by Hamada et al. is devoted to simulation of a dosage optimization of two antibiotics, cefmetazole and flomoxef. The approach described in the study is very interesting, but seems to be quite specific. As a conclusion, the authors recommend a new regimen of the two antibiotics, but their recommendation is to be validated in clinical trials which is also declared by the authors.
I have following comments on the submission:
1) Page 1, line 38: a whitespace is missing
2) Page 2, Section 2, lines 66-87:
This huge paragraph contains a lot of data on the regimen of the antibiotics under study, their PK at different dosages and ways of administration. Firstly, it is not clear, what was the source of all these data - literature or in-house observations. Secondly, this paragraph needs a mini-introduction which would prepare a reader to the information following. And this paragraph, in my opinion, would look better if divided into two paragraphs describing the two antibiotics and starting from "Figure 1-A" and "Figure 1-B." correspondingly.
The main text does not indicate that the PTA curves were simulated, and how it was done. From the section "Materials and methods" it is not clear, how they were simulated.
And finishing this comment with one which would be better to start: what does the PTA parameter represent?
3) Page 4, line 136: The abbreviation TAM should be explained here.
4) Page 6, line 210: Should it be there the word "utilized" instead of "analyzed"?
5) Here, Table 2: The three-level fractions in the expressions look not good.
6) Conclusions (or somewhere else where appropriate): Many antibiotics are monitored in patients during treatment. Judging by the frequency of using cefmetazole and flomoxef, these drugs are candidates for monitoring. As I could see, no data on this approach can be found in literature. So, could the authors reflect this to highlight the importance of their study?
Author Response
Manuscript ID: antibiotics-1628673
Title: Pharmacokinetic/ pharmacodynamic analysis and dose optimization of
cefmetazole and flomoxef against extended-spectrum β-lactamase-producing
Enterobacterales in patients with invasive urinary tract infection considering renal function
Type of manuscript: Article
Journal: Antibiotics
Thank you for your reviewing our article. We made some changes in our manuscript according to reviewer’s suggestions with yellow highlight. We think some revises enhanced the quality of our manuscript.
Comments and Suggestions for Authors
The article submitted by Hamada et al. is devoted to simulation of a dosage optimization of two antibiotics, cefmetazole and flomoxef. The approach described in the study is very interesting, but seems to be quite specific. As a conclusion, the authors recommend a new regimen of the two antibiotics, but their recommendation is to be validated in clinical trials which is also declared by the authors.
I have following comments on the submission:
- Page 1, line 38: a whitespace is missing
Response: Thank you for your comment. We added the Space.
2) Page 2, Section 2, lines 66-87:
This huge paragraph contains a lot of data on the regimen of the antibiotics under study, their PK at different dosages and ways of administration. Firstly, it is not clear, what was the source of all these data - literature or in-house observations. Secondly, this paragraph needs a mini-introduction which would prepare a reader to the information following. And this paragraph, in my opinion, would look better if divided into two paragraphs describing the two antibiotics and starting from "Figure 1-A" and "Figure 1-B." correspondingly.
Response: Thank you for your advice. I have separated the paragraphs.
The main text does not indicate that the PTA curves were simulated, and how it was done. From the section "Materials and methods" it is not clear, how they were simulated.
Response: Thank you for your comment. We added “The clinical breakpoints were calculated to attain PTA≥90% for creatinine clearance (CCR) of 10, 30, 50, and 70 mL/min, respectively. The 90%≥PTA (70% T> MIC) of cefmetazole and flomoxef in patients who received a standard regimen (0.5 or 1 g, 1 hour injection) for each renal function was calculated.”
And finishing this comment with one which would be better to start: what does the PTA parameter represent?
Response: Thank you for your comment. We added “The probability of target attainment (PTA) was calculated by simulation using dosage, renal function, and MIC for cefmetazole and flomoxef, respectively. Table 3 was used for PK parameters [35, 36]. The PTA was calculated as the probability of 70% T>MIC and the results are shown in Figure 1.”
Page 4, line 136: The abbreviation TAM should be explained here.
Response: Thank you for your comment. We added “the information on time above minimal inhibitory concentration (TAM), a PK/PD parameter, was not available.”
4) Page 6, line 210: Should it be there the word "utilized" instead of "analyzed"?
Response: Thank you for your comment. We changed to "utilized".
- Here, Table 2: The three-level fractions in the expressions look not good.
Response: Thank you for your comment. Changed it.
- Conclusions (or somewhere else where appropriate): Many antibiotics are monitored in patients during treatment. Judging by the frequency of using cefmetazole and flomoxef, these drugs are candidates for monitoring. As I could see, no data on this approach can be found in literature. So, could the authors reflect this to highlight the importance of their study?
Response: Thanks for the important point. I have added the following. To our knowledge, appropriate doses of these drugs for ESBL‐producing Enterobacterales have not been studied according to renal function.

Reviewer 2 Report
Thank you for this work. Useful study on PK/PD of Cefmetazole and Flomoxef. The manuscript is well written and provides clear information on dosing schedules.
I have no major recommendations
I wonder if there are avialable data for Paeds
Author Response
Thank you very much for your peer review.
Currently we do not have pediatric data, but we would like to consider pediatric data in the future.
Reviewer 3 Report
Please see the attachment.

Author Response
Manuscript ID: antibiotics-1628673
Title: Pharmacokinetic/ pharmacodynamic analysis and dose optimization of
cefmetazole and flomoxef against extended-spectrum β-lactamase-producing
Enterobacterales in patients with invasive urinary tract infection considering renal function
Type of manuscript: Article
Journal: Antibiotics
Thank you for your reviewing our article. We made some changes in our manuscript according to reviewer’s suggestions with yellow highlight. We think some revises enhanced the quality of our manuscript.
The manuscript sheds light upon PK/PD of essential antibiotics for treatment of UTIs caused by difficult to treat ESBL producing Enterobacterales. The findings are interesting and quite useful for dose optimization and usage of both antibiotics in clinical setting. However, some critiques need to be addressed before publication.
-The authors must confirm they have obtained ethical approval from an institutional or local ethics review board.
Response: I have added the following. The study was approved by the institutional review board (No. NCGM-G-004083-00).
-As the study involved patients with different degrees of renal failure therefore, I can’t find that the manuscript title conveys this critical point.
Response: Thanks for your point important. I have added considering renal function to the title.
Abstract:
-The authors need to show what did this study add to what was already known on this topic?
Response: Thanks for your pertinent points. I have added the following.
Monte Carlo simulation (MCS) was performed using the published data to calculate a new probability of target attainment (PTA≥90%) for each renal function.
Introduction:
-Re-writing to some parts of introduction is needed as there is no cohesion/coherence in some parts e.g., lines 39-41.
Response: Thank you for your comment. We have corrected it.
-Strike off [considered to be] line 35, same line add [development of] before [ antimicrobial-resistant bacteria]
Response: We have corrected it.
- Space off [ byextended-spectrum β-lactamase] in line 38.
Response: Thank you for your comment. We added the Space.
-There is no such thing called [ESBLPE], only ESBL used.
Response: Changed it all to ESBL‐producing Enterobacterales
-Something missed in line 40 before [development], please revise.
Response: Thank you very much. Added it.
-43-47, more references are needed.
Response: Thank you very much. Added it.
-Define [MERINO].
Response: We added, The RCT Meropenem vs Piperacillin-Tazobactam for Definitive Treatment of BSI's Due to Ceftriaxone Non-susceptible Escherichia Coli and Klebsiella Spp. (MERINO)
-Line 50 italicize [in vitro].
Response: Changed it.
-Remove [unfortunately], line 55.
Response: Delete it.
-Delete [are scarce] line 57.
Response: Delete it.
Results:
-Quality of figures are poor. Figure 1A and 1B, the letters A and B would be tidy more if you put them on the top left-hand corner.
Response: Thanks for your point important. We have improved the data to the cleanest data we can now.
Discussion:
-Line 130, reference is required.
Response: Added it.
-Line 141, remove [s] from [ binding s].
Response: Delete it.
-Lines 136, 137, 144, 145, 147 and so on, define [TAM].
Response: Thank you for your comment. We added first TAM at L 136, “the information on time above minimal inhibitory concentration (TAM), a PK/PD parameter, was not available.”
